# Does Neuraxial Anesthesia as General Anesthesia Damage DNA? A Pilot Study in Patients Undergoing Orthopedic Traumatological Surgery

**DOI:** 10.3390/ijms21010084

**Published:** 2019-12-20

**Authors:** Monika Kucharova, David Astapenko, Veronika Zubanova, Maria Koscakova, Rudolf Stetina, Zdenek Zadak, Miloslav Hronek

**Affiliations:** 1Department of Biophysics and Physical Chemistry, Faculty of Pharmacy, Charles University, Heyrovskeho 1203, 500 05 Hradec Kralove, Czech Republic; kucharova@faf.cuni.cz (M.K.); Veronikazubanova@gmail.com (V.Z.); makoscakova@gmail.com (M.K.); 2Department of Research and Development, University Hospital in Hradec Kralove, Sokolska 581, 500 05 Hradec Kralove, Czech Republic; david.astapenko@fnhk.cz (D.A.); r.stetina@tiscali.cz (R.S.); zdenek.zadak@fnhk.cz (Z.Z.); 3Department of Anesthesiology, Resuscitation and Intensive Medicine, University Hospital Hradec Kralove, Sokolska 581, 500 05 Hradec Kralove, Czech Republic; 4Institute for Clinical Biochemistry and Diagnostics, Charles University, Faculty of Medicine in Hradec Kralove and University Hospital Hradec Kralove, Sokolska 581, 500 05 Hradec Kralove, Czech Republic; 5Department of Biological and Medical Sciences, Faculty of Pharmacy, Charles University, Zborovska 2089, 500 03 Hradec Kralove, Czech Republic

**Keywords:** DNA damage, general anesthesia, neuraxial anesthesia, comet assay

## Abstract

The human organism is exposed daily to many endogenous and exogenous substances that are the source of oxidative damage. Oxidative damage is one of the most frequent types of cell component damage, leading to oxidation of lipids, proteins, and the DNA molecule. The predominance of these damaging processes may later be responsible for human diseases such as cancer, neurodegenerative disease, or heart failure. Anesthetics undoubtedly belong to the group of substances harming DNA integrity. The goal of this pilot study is to evaluate the range of DNA damage by general and neuraxial spinal anesthesia in two groups of patients undergoing orthopedic traumatological surgery. Each group contained 20 patients, and blood samples were collected before and after anesthesia; the degree of DNA damage was evaluated by the comet assay method. Our results suggest that general anesthesia can cause statistically significant damage to the DNA of patients, whereas neuraxial anesthesia has no negative influence.

## 1. Introduction

DNA is continuously exposed to a variety of biological, chemical, and physical agents, which may alter its structure and modify its function. Such exogenous compounds include anesthetic substances commonly used in general anesthesia (GA) or neuraxial anesthesia (NA). These have attracted attention because of concern about their potential genotoxic effect. The aim of anesthesia during surgery is to reduce pain. However, anesthetics cannot completely block autonomic nervous system responses to traumatic stimuli. The autonomic nervous system-dependent responses to traumatic stress remain progressive, resulting in increased oxygen consumption and production of oxygen free radicals, and decreased antioxidant activity. Typically, several different drugs and techniques are combined during anesthesia [1]. Various methods have described DNA structure damage due to anesthesia [2,3,4,5], and thus there is evidence that anesthetics change human genetic information. Recent studies have been published concerning the genotoxicity of inhalational anesthesia in patients who have undergone surgery and in personnel who are occupationally exposed to anesthetics. Even a trace concentration of waste anesthetic gases may lead to an increase in genetic damage [6,7].

In GA, inhalation anesthetics are the most widely used. The anesthetic mechanism of volatile agents is complicated, targeting a number of sites including GABA (gamma aminobutyric acid) and NMDA (N-methyl-D-aspartic acid) receptors. Chemically they are small hydrophobic molecules that pass through lipophilic cell membranes. They cause depression of respiration and oxidative phosphorylation in mitochondria. Exposure to volatile anesthetics generates small quantities of reactive oxygen species (ROS) either directly due to interaction with the electron transport chain or indirectly through a signaling cascade in which G-protein-coupled receptors, protein kinases, and mitochondrial ATP-sensitive potassium channels are involved [8].

Among the inhaled anesthetic gases, the halogenated gases isoflurane (ISF) and sevoflurane (SVF) are the most widely used in general anesthesia. ISF was synthesized in 1965 and introduced to clinical practice several years later. According to the study by Corbett from 1976, this anesthetic may cause liver tumors in rats [9]. Even though this claim was not confirmed in other studies [10], results are not entirely uniform. The chemical structure of ISF is similar to that of some non-anesthetic carcinogens, including chloromethyl methyl ether [11]. The advantage of ISF and SVF is their low metabolism rate and low blood–gas partition coefficient, which decreases its induction and recovery times.

Some studies have shown an association between inhalational anesthesia and increased risk of tumor spread [12,13]. In their retrospective study, Wigmore and coauthors found that cancer patients had a worse survival outcome if they received inhalation anesthesia. Inhalation anesthesia inhibits the immune system by diminishing the function of killer cells that protect the organism against the proliferation of cancer cells [14]. It has been published that ISF could promote the growth and migration of glioblastoma cells and increase levels of hypoxia-inducible factors that are overexpressed in a variety of carcinomas and their metastases, and it is supposed that it is a transcriptional regulator of VEGF expression involved in tumor growth [15,16]. A connection with postoperative cognitive impairment and possible development of Alzheimer’s disease is suspected rather than proved [17,18]. 

The need for a local anesthetic with low toxicity has led to the development of numerous compounds. Bupivacaine, levobupivacaine, and ropivacaine are long-acting amide-based local anesthetics most commonly used in clinical practice. The principle of their effect is the prevention of nerve impulse induction primarily in nerve cell membranes by inhibition of voltage-gated Na^+^ channels [19]. Regional analgesic techniques deliver local anesthetics to groups of peripheral nerves outside the central nervous system. They block sensation to specific dermatomes. The principle of action of neuraxial techniques is to deliver local anesthetic and analgesic medications directly or indirectly to the spinal cord [20]. 

Neuraxial blockade includes epidural and subarachnoid (spinal) anesthesia (SA). The difference between them is in the dosage and the place of administration. In this manner, the speed of the onset of the numbness and the speed of the influencing of motoric is given. During epidural anesthesia, spinal nerve root block occurs by injection of a local anesthetic outside the subarachnoid space. The required dose and concentration of the local anesthetic are high. Efficacy starts slowly, and the duration of action is medium to long (6–8 h). It has a wide range of applications including operative anesthesia, obstetric analgesia, and chronic pain management. Spinal anesthesia is the oldest neuraxial technique, first used by August Bier who injected cocaine into the intrathecal space to provide anesthesia in 1898. During SA, the spinal nerve roots are blocked by the injection of local anesthetic to the subarachnoid space. The concentration of local anesthetic is usually high, but the volume required is usually small. Efficacy starts very quickly, sometimes during the injection. The duration of action depends on the substance used: it may be short, medium, or long (minutes to hours). Use of this technique in general is confined to shorter procedures to the lower extremities or pelvis [20,21]. As with any anesthesia, subarachnoid blockade carries the risk of adverse reactions, such as post-puncture headaches, hypotension, bradycardia, hypothermia, urinary retention, nausea, and vomiting [21].

Genotoxic or mutagenic effects of anesthesia have been of interest in many studies. However, the results of the studies do not coincide, and the differences may be influenced by many factors. Different exposure times, interindividual differences in sensitivity dependent on genetic factors, different design of experiment, type and length of surgery, interindividual parameters of patients, and dissimilar methods of statistical analysis all contribute to the differences in analytical results [22]. DNA damage during GA has been well documented, and it delivers a significant burden to the patients [23,24,25]. Data on NA are less common. There are several studies that prove that local anesthetics cause neurotoxicity and apoptosis by induction of oxidative DNA damage [26,27,28]. Unfortunately, there is a lack of information about the influence of spinal anesthesia on DNA damage. Which method offers less burden for the organism remains controversial. Therefore, there is a tendency to continually explore and improve its safety for humans [4].

A suitable method for quantification of the degree of DNA damage in human medicine is the comet assay (single cell gel electrophoresis) [29]. The method was developed in a different modification enabling quantification of single-strand DNA breaks (SSB), double-strand DNA breaks, and oxidized pyrimidine and purine bases, and enables quantification of low-level DNA changes in individual eukaryotic cells [30]. Using this method, the DNA changes in peripheral lymphocytes can be evaluated. Comet assay is appropriate for quantification of DNA damage in patients especially after chemotherapy and radiotherapy [31,32].

The primary objective of this pilot study is to evaluate the applicability of the comet assay method for quantification of DNA changes in patients under anesthesia undergoing orthopedic or traumatological lower limb surgery. The secondary aim is to verify the hypothesis that neuraxial anesthesia damages DNA less than general anesthesia. Obtaining such data should contribute to broadening the knowledge on the impact of surgical interventions on oxidative DNA damage, and should be the basis for further research into finding preventive and/ or protective procedures to minimize DNA damage associated with invasive interventional procedures in clinical medicine.

## 2. Results

### 2.1. Preoperative Data

Forty-five patients were included in the study, from 40 of whom were obtained complete data. Twenty patients were in the general anesthesia group (GA group) and 20 in the spinal anesthesia group (SA group). The demographic data and results of the preoperative evaluation are in Table 1. The level of statistical significance was *p* = 0.05. Both groups were comparable except for age and height. Patients were separated into four groups according to ASA (American Society of Anesthesia) classification: ASA I represents normal, healthy patients; ASA II includes patients with mild systemic disease; ASA III includes patients with severe systemic disease; ASA IV includes patients with severe systemic disease that is a constant threat to life. The differences between patients in individual ASA grades between the GA group and the SA group were statistically insignificant (*p* = 0.49).

Relevant laboratory results prior to anesthesia and preoperative performance data are shown in Table 2. Patients in GA had a statistically significantly higher duration of operation. 

### 2.2. DNA Damage

The values of DNA damage for both groups are shown in Table 3. The determined parameter was the percentage of DNA in the tail of the comet. In the table are the percentage values of DNA SSB, oxidized pyrimidine bases (ENDO III), and oxidized purine bases (FPG). In the general anesthesia group, the results show a statistically significant difference between the before and after blood samples for all three parameters (for SSB *r* = 0.84. *p* < 0.0001; for ENDO III *r* = 0.83. *p* < 0.0001; for FPG *r* = 0.72. *p* < 0.0001; Table 3, Figure 1).

The results of spinal anesthesia showed statistically insignificant differences between before and after samples for all the monitored parameters (for SSB *p* = 0.97; for ENDO III *p* = 0.29; for FPG *p* = 0.41; Table 3, Figure 2).

### 2.3. The Influence of Length of Anesthesia

The dependence between the duration of anesthesia and SSB breaks in the general anesthesia group came out as statistically significant. This correlation is expressed as linear dependence with *R^2^* = 0.36 (Figure 3). In the spinal anesthesia group, such dependence has not been described (Figure 4).

## 3. Discussion

Many drugs and environmental factors can cause DNA damage. Thus, it is important to understand and be able to predict correctly the effect of the DNA damaging agent. The ultimate biological effect of exposure depends on many processes within the cell that affect DNA damage and repair [33]. It is also known that the level of normal oxidative damage is different between individuals according to the condition of antioxidative and stress response systems. 

The aim of our pilot study is to describe DNA damage due to the different types of anesthesia during surgery of a similar range. Many authors have addressed the case of general anesthesia. Sardas and co-authors described DNA damage in patients after abdominal surgery under general anesthesia with isoflurane. They discovered a statistically significant difference between the experimental and control group and between states before and after anesthesia [5]. Similar results were obtained by Karybiyik et al. They compared a group of 24 un-premedicated patients with ASA grades 1–2 with a control group consisting of 12 healthy individuals. They used the comet assay method and recorded a significant increase in the mean comet response in blood sampled from patients at 60 and 120 min of anesthesia, and on the first day after anesthesia, in both sevoflurane and isoflurane treated groups [23]. Kadioglu et al. dealt with the influence of sevoflurane on DNA damage in patients indicated for mastectomy. They used two methods—the comet assay and the alkaline halo assay. They described statistically significant damage to DNA by both methods after two-hour surgery [34]. Nogueira and et al. described statistically significant damage the day after minor surgery under general anesthesia maintained with desflurane [35]. Braz with coworkers assessed the DNA damage in patients who had undergone minimally invasive surgery lasting two hours under inhalation of sevoflurane and propofol. No significant difference in DNA damage was observed in either group of patients, and their study is unique [4]. The authors of the cited studies followed the state of the DNA on the several postoperative days. They agree that full DNA repair will occur between the 3rd and 5th postoperative day [5,23,34].

The results of the above studies together with our results raises the questions about the genotoxicity of anesthetics causing iatrogenic damage. Anesthetics, as highly lipophilic substances, do not need any specific transporter to get inside the cell nucleus. It is known that the potential toxic degradation product of SVF is fluoromethyl 2-2 difluoro-1-(trifluoromethyl)vinyl ether (Compound A). The highly reactive nature of Compound A suggests that it may be an alkylating agent and is well known that alkylating agents are electrophilic compounds with affinity towards the nucleophilic parts of macromolecules [23].

Anesthesia as a chemical agent belongs to the group of intracellular agents that cause DNA changes due to its participation in the production of reactive oxygen species (ROS). ROS overproduction leads to oxidative changes on membranes, proteins (also histones), or DNA in the form of DNA breaks. This condition disrupts the homeostasis of the cell and, in combination with the disruption of the mitochondrial transmembrane potential caused by the presence of a pro-oxidant status, could accelerate apoptosis of lymphocytes. We also think that on the other hand, chemical agents could also participate in chemical modification of DNA bases that can be checked by enzymatic modification of the comet assay. These changes lead to mistakes in base pairing and could result in mutagenesis [36].

In our pilot study, two blood samples from each of the 40 patients were processed and the degree of DNA damage was evaluated by the comet assay method. The results showed a statistically significant increase in DNA damage in all three monitored parameters (SSB, and oxidized pyrimidine and purine bases) after surgery in the 20 patients undergoing general anesthesia. In the spinal anesthesia group, no statistically significant differences were found. This study has several limitations. We are aware that both groups of patients were disparate, in the type of surgery resulting from diagnosis, in that the surgery under general anesthesia required more time, and in that some patients underwent repeated surgery (for example, patients after polytrauma). Also, the age of the patients in both groups was statistically significantly different; the patients receiving spinal anesthesia being appreciably older. This was driven by the type of diagnosis, with older patients undergoing knee or hip replacement with total arthroplasty. Our results show that there is a linear dependence between the duration of general anesthesia and the level of single-strand breaks (Figure 3). This dependence has not been demonstrated in spinal anesthesia (Figure 4).

Our aim is to confirm the applicability of comet assay in the research of DNA damage caused by different types of anesthesia. This assay is now widely accepted as a standard method for assessing DNA damage in individual cells and is used in a broad variety of applications including human biomonitoring, genotoxicology, ecological monitoring, and as a tool to investigate DNA damage and repair in different cell types in response to a range of DNA-damaging agents. Because there is no uniform protocol for comet assay, for every lab, the verification of its own protocol is needed. Our protocol was tested and optimized during previous research, where the range for positive and negative results was found [37]. For this experiment, we did not have a positive control yet, but for the next experiment, the samples containing a higher level of oxidative damage should be used as a positive control.

The aim of future experiments should be to expand the patient group to validate the results and verify that these findings will apply to larger experimental groups. It will be necessary to repeat the measurements with larger and more homogeneous groups of patients and collect more samples to have a good statistical strength (to have few age groups, groups according to the duration of anesthesia) where the effect of anesthesia according to higher age or longer duration of anesthesia should be evaluated. It should also be interesting to include some patients with the combination of anesthesia and another oxidative agent (smoking, chemotherapy) to search for the effect of combined agents. A good idea should be the inclusion of other methods for evaluation of biological levels of basic defense or antioxidative system compounds (enzyme activity, protein, or mRNA levels) to compare the interindividual differences in the ability to fight against harmful agents.

Our work has shown that a modified alkaline version of the comet analysis using enzymes is suitable for the quantification of single-strand breaks and oxidized purines and pyrimidines in the DNA of lymphocytes. The advantage of the method is the small number of cells required for analysis, its sensitivity, and the detection of single-cell damage. It is a quick and relatively simple method for assessing oxidative damage. The method is suitable for clinical trials as it does not present a burden on the patient when taking blood samples or by exposure to harmful substances.

This study is a pilot study whose primary objective was to verify the feasibility of the method and with the secondary goal of verifying the hypothesis that neuraxial spinal anesthesia causes less damage to DNA than general anesthesia. Our results and results of the cited studies on patients undergoing surgery and on operating room staff [6,36] raise further questions for the prevention of DNA damage in both patients and operating room staff [38]. Notwithstanding, it is also difficult to assess the impact of surgical trauma on DNA integrity; we may assume it plays a minor role, as DNA damage after surgery was minimal in the SA group.

## 4. Materials and Methods

### 4.1. Selection of Patients

This prospective, monocentric, non-randomized study was approved by the Ethics Committee (University Hospital Hradec Králové. reference number 201511 S14P; identification date 22.10.2015.). All patients signed informed consent to participate in the study. The Helsinki Declaration of Patients’ Rights was respected. The inclusion criteria were consent to the study and signing of an informed consent form, age over 18, traumatological major surgery of the lower limb or pelvis under general anesthesia, or orthopedic major joint replacement lower limb surgery under neuraxial anesthesia. The exclusion criteria were acute nature of the procedure, a history of cancer with chemotherapy or radiotherapy in the last 12 months, immunosuppressive therapy in the last 12 months, active smoker (more than 1 cigarette/ day), a CT scan in the last week before surgery, and patient refusal to participate. Forty-five patients participated in the study and complete data were obtained from 40 of them. The age of patients in the GA group was between 18 and 66 years (40 ± 15.6 years), with weight in the range 60 to 120 kg (88.5 ± 170.3 kg) and height between 164 and 189 cm (176.3 ± 7.5 cm); BMI was between 22.04 and 40.68 (28.4 ± 5.2). Patients in the SA group were aged between 18 and 88 years (62 ± 15.3 years), with weight in the range 66 to 115 kg (81.2 ± 14.0 kg) and height between 150 and 190 cm (170 ± 9.2 cm); BMI was between 21.74 and 37.55 (27.6 ± 4.4).

### 4.2. Anesthetic Management

The group of patients taking general anesthesia (GA) underwent traumatological surgery in the lower limb or pelvis (open reduction with internal fixation, so-called ORIF). Patients received premedication of 1.5 mg bromazepam (Lexaurin, Kabu Pharma, Prague, Czech Republic) per os. GA was induced by propofol (Propofol, Fresenius Kabi, Bad Homburg, Germany) at a dose of 2 mg/kg. Analgesia was administered with sufentanil (Sufentanil Torrex, Chiesi Pharmaceuticals, Vienna, Austria) at an initial dose of 10 µg and further in accordance with monitored SPI (surgical plethysmography index). If muscle relaxation was required, atracurium (Tracrium, Aspen Pharma, Dublin, Ireland) was used at an initial dose of 0.5 mg/kg followed by monitoring of the depth of muscle relaxation (the monitor was an integral part of the Aisys anesthesia machine) to TOF 2 (train of four). GA was maintained by isoflurane (Forane, Abott Laboratories Ltd., Maidenhead, UK) in a carrier mixture of oxygen (FiO_2_ 0.40) with nitrous oxide with a minimum alveolar concentration of 0.8–1.0. If decurarization was required at the end of the procedure, neostigmine (Syntostigmine, BB Pharma, Prague, Czech Republic) was administered 1.5 mg intravenously and atropine (Atropine Biotika, BB Pharma, Prague, Czech Republic) 0.5 mg intravenously. The demographic data of each patient was recorded and blood pressure, pulse, peripheral blood oxygen saturation, exhaled carbon dioxide concentration, and body temperature measured every 5 min during the surgical procedure. Fluid therapy was maintained with a balanced crystalloid solution (Ringerfundin, BBraun, Melsungen, Germany) via a peripheral venous catheter (18 G, BBraun, Melsungen, Germany) at a baseline rate of 100 mL/h and according to blood loss and circulatory stability. Body temperature was maintained in a physiological range between 36 and 37 °C with a heating pad (Astopad, Stihler electronic, Stuttgart, Germany) and an air heater (Warm Air, Polymed, Hradec Kralove, Czech Republic). Airways were secured by orotracheal intubation (SUMI, Sulejowek, Poland) or laryngeal mask (Teleflex, Dublin Road, Athlone, Westmeath, Ireland). Artificial lung ventilation was volume-controlled or pressure-controlled-volume-guaranteed to a tidal volume of 6 mL/kg with a respiratory rate of 12–16 breaths/minute (according to the exhaled concentration of carbon dioxide range between 35–45 mmHg) and a positive end expiratory pressure 4 cm H_2_O. The Aisys anesthesiology machine (GE Healthcare. Prague, Czech Republic) was used.

The group of patients with neuraxial spinal anesthesia (SA) underwent orthopedic surgery—knee or hip replacement with total arthroplasty under subarachnoid anesthesia. Subarachnoid puncture was performed aseptically by Quinke needle (25G 88 mm, BBraun, Melsungen, Germany) in the L2/3 or L3/4 intervertebral space by a paramedial approach after a previous infiltration of the injection site with 1% mesocaine (Mesocaine, Zentiva, Prague). 2.2 mL of 0.5% levobupivacaine (Chirocaine, AbbVie, Prague, Czech Republic) and 2.5 µg of sufentanil (off-label administration; Sufentanil Torrex, Chiesi Pharmaceuticals, Vienna, Austria) were administered intrathecally. Circulatory instability following induction of subarachnoid anesthesia was resolved by titration with 10 mg doses of ephedrine (Ephedrin Biotika, BB Pharma, Prague, Czech Republic). During the procedure, patients received sedation with midazolam (Midazolam Accord, Accord Healthcare Polska, Warsaw, Poland). Patients were given oxygen by face mask at a supply of 5 L/min when the saturation dropped below 93%. Monitoring and data recording were similar to the GA group.

### 4.3. Comet Assay

#### 4.3.1. The Lymphocyte Isolation

Four mL blood samples were obtained from patients before and immediately after the surgery into sodium citrate tubes and then were subjected to DNA damage analysis using the comet assay method previously described [39,40,41]. Briefly, blood was carefully laid over 4 mL LSM (Biotech, Austria) and tubes were centrifuged (30 min, 1500 rpm, 20 °C). The formed ring of lymphocytes (Figure 5) was transferred to centrifuge tubes (Sarstedt, Austria) and 10 mL of PBS (Sigma Aldrich, Saint Louis, MO, USA) was added. The concentration of cells in the sample was calculated using a Bürker chamber, and samples were centrifuged again (10 min, 1500 rpm, 8 °C). The sediment was re-suspended with PBS buffer (Sigma Aldrich, Saint Louis, MO, USA) and adjusted to a concentration of 1 million cells/mL.

#### 4.3.2. Alkaline Version of Comet Assay

The alkaline modification of the comet assay was used for the determination of DNA damage [40]. For observation of individual types of damage (SSB breaks and oxidized pyrimidines and purines), lymphocytes were pipetted in 35 µL portions to the 1.5 mL microtubes (Eppendorf, Hamburg, Germany). To a microscope slide pre-coated with 1% aqueous agarose for electrophoresis (Serva, Heidelberg, Germany) was added 85 µL of a 1% solution of high melting point agarose in PBS (HMP agarose, Sigma Aldrich, Saint Louis, MO, USA), and the gel left at 4 °C to allow solidification. A mixture of the 35 µL cell suspension (approximately 35000 cells) and 85 µL of 1% low melting point agarose in PBS (LMP agarose, Sigma Aldrich, Saint Louis, MO, USA) was spread onto this high melting point agarose and again left at 4 °C to allow solidification.

#### 4.3.3. Cell Lysis

Once the top layer had solidified, the slide was gently immersed in cold lysing solution (2.5 M NaCl, 100 mM EDTA, and 10 mM Tris-HCl pH 10 to which 1% Triton X-100 and 10% DMSO had been freshly added). The slides were left at 4 °C for at least 1 h.

#### 4.3.4. Enzymatic Digestion

The slides which were used for enzymatic digestion were gently removed from the lysing solution, washed three times with ENDO buffer (0.1 M KCl, 40 mM HEPES, 0.5 mM EDTA, and 200 µL/mL BSA, pH 8, 37 °C), and 30 µL of the specific enzyme was added. We used the specific DNA endonuclease III (ENDO III) to detect oxidized pyrimidines and the FPG (formamidopyrimidine -DNA glycosylase) to identify altered purines. The slides were covered with coverslips and incubated in a thermobox at 37 °C for 1 h.

#### 4.3.5. Unwinding, Electrophoresis, and Staining

The coverslips were removed from the incubated slides. All slides (incubated and unincubated) were placed in a horizontal electrophoresis tank (Model A5, Owl separation systems, Inc., Thermo Scientific, Waltham, MA, USA) filled with fresh electrophoresis buffer (300 mM NaOH and 1 mM EDTA, pH 13). The slides had to be covered with the buffer and exposed to alkali for 40 min to allow DNA unwinding and cleavage of alkali-labile sites. Electrophoresis was then conducted at 33 V, 300 mA for 30 min at 4 °C by using an electrophoresis power supply EPS 300 IIV (C.B.S. Scientific Company, Inc., CA, USA). The negatively charged DNA bases migrate to the positively charged anode and form the shape of a comet when subjected to an electric field. The size and the shape of the comet and the distribution of the DNA within the comet correlate with the extent of DNA damage. After electrophoresis, the slides were removed from the tank and washed three times for 5 min with neutralizing buffer (0.4 M Tris-HCl, pH 7.5) and finally with distilled water. Excess liquid was blotted from each slide and the DNA stained with 20 µL ethidium bromide. A clean coverslip was then placed over the slide.

#### 4.3.6. DNA Damage Evaluation

Slides were evaluated under a fluorescent microscope with an optical system (NIKON INSTRUMENTS INC., Melville, NY, USA). One hundred cells per slide were scored according to % tail DNA by the software LUCIA Comet Assay (Laboratory Imaging, Prague, Czech Republic). This means a total of 300 cells were analyzed per subject. The DNA analysis is expressed as the ratio of the DNA intensity in the tail relative to the head of the comet. The percentage tail DNA was determined for single-strand breaks in the DNA (SSB), pyrimidine damage (ENDO III), and purine damage (FPG). After evaluation, the slides were washed and allowed to dry.

#### 4.3.7. Statistical Evaluation

The acquired data were analyzed using the programs Graph-Pad Prism7 (GraphPad Software, La Jolla, CA, USA) and Excel 2016 (Microsoft, Redmont, WA, USA).

The normality distribution of the demographic data and the results of the laboratory tests was demonstrated by D’Agostino and Pearson test and the Shapiro–Wilk normality test. According to the normality of the distribution, the Mann–Whitney test was used for comparison of the demographic data and laboratory tests of the two experimental groups. It is used to evaluate unpaired experiments when comparing two sample files or two experimental reaches. It is used when due to the small number of the experimental data or their nature, there is uncertainty about the normality of the distribution. Results were expressed as the medians and first and third quartiles (25% percentile, 75% percentile).

For evaluation of the statistical significance of the DNA damage between the general anesthesia and spinal anesthesia groups, the Wilcoxon matched-pairs signed-rank test was used. It tests the match of two medians and does not assume normality of their distribution. The differences between patients in individual ASA grades between the GA group and SA group were tested by the Mann–Whitney test.

## Figures and Tables

**Figure 1 ijms-21-00084-f001:**
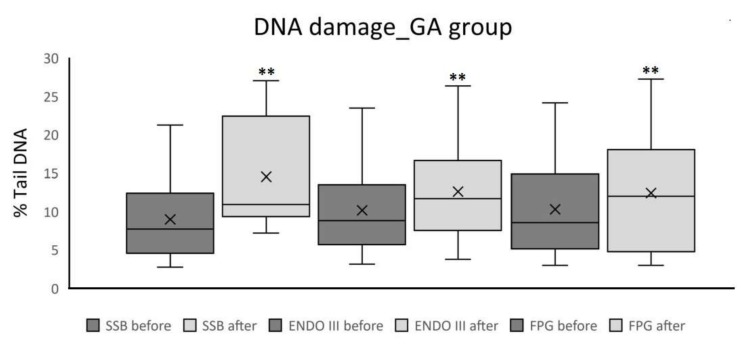
Percentage rail DNA as a measure of SSBs, oxidized pyrimidine bases, and oxidized purine bases in patients underwent general anesthesia (median ± first and third quartile, whiskers are minimal and maximal value), ** *p* < 0.0001.

**Figure 2 ijms-21-00084-f002:**
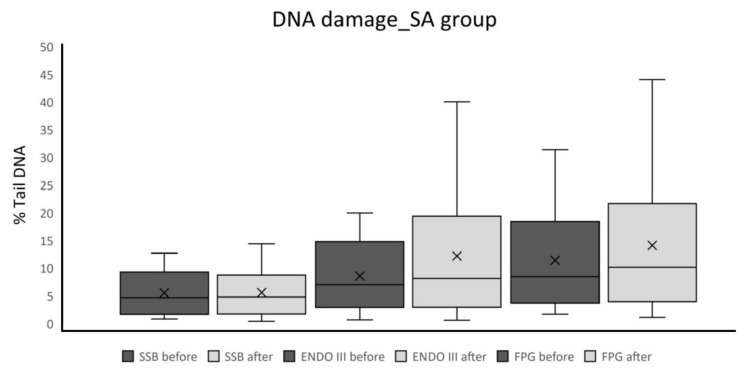
Percentage Tail DNA as a measure of SSBs, oxidized pyrimidine bases, and oxidized purine bases in patients who underwent spinal anesthesia (median ± first and third quartile, whiskers are minimal and maximal value).

**Figure 3 ijms-21-00084-f003:**
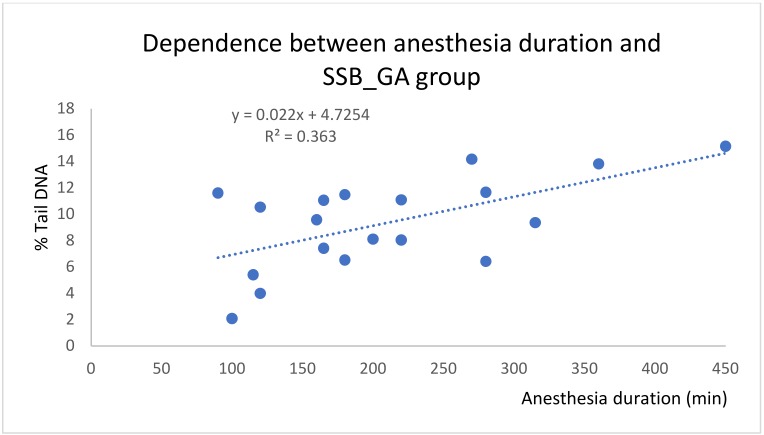
Linear dependence between anesthesia duration and SSB for the general anesthesia group.

**Figure 4 ijms-21-00084-f004:**
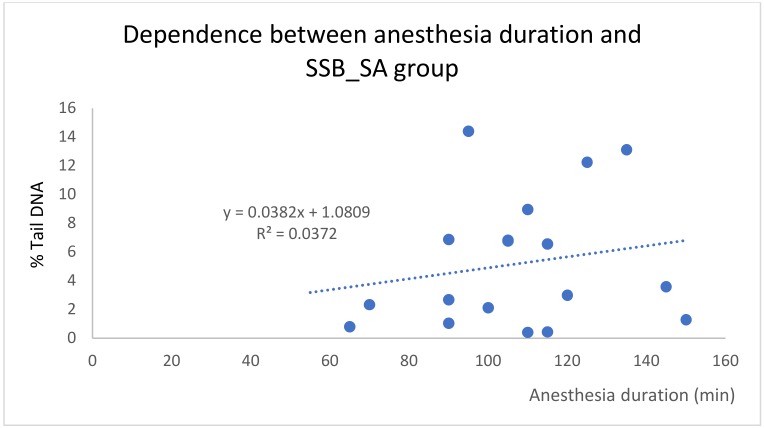
Dependence between anesthesia duration and SSB for the spinal anesthesia group.

**Figure 5 ijms-21-00084-f005:**
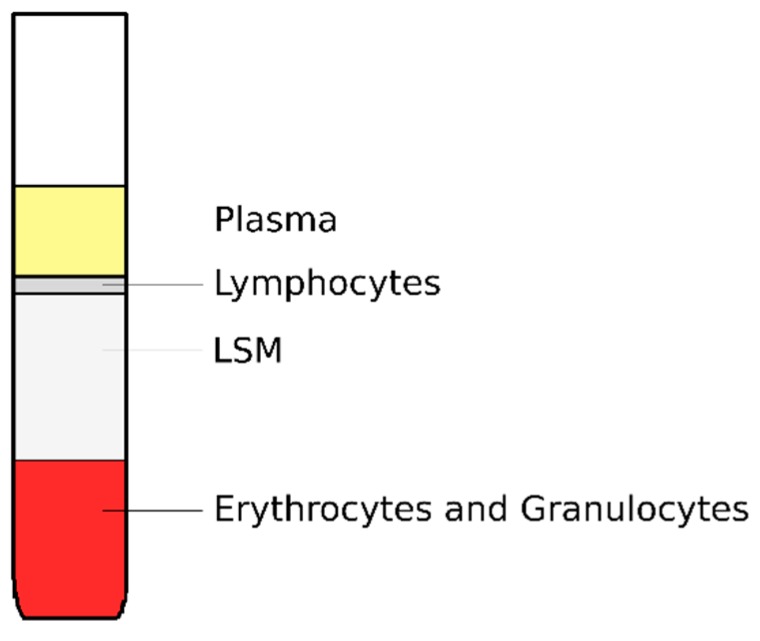
The separation of blood elements using PBS

**Table 1 ijms-21-00084-t001:** Demographic data and results of preoperative evaluation of both groups.

Parameter	GA Group	SA Group	*p*-Value
Number	20	20	
Woman	9	11	0.96
Man	10	10	0.96
Age [years]	37 (29; 51)	65.5 (53.25; 74.75)	0.0002 *
Height [cm]	175.5 (170; 183)	170 (166; 175)	0.01 *
Weight [kg]	90 (80; 99)	81,2 (73.5; 85)	0.12
BMI	28 (24; 30)	28.4 (26; 30)	0.64
ASA I [%]	13	21	0.55
ASA II [%]	54	68	0.36
ASA III [%]	27	11	0.22
ASA IV [%]	6	0	0.25

BMI—body mass index; ASA—American Society of Anesthesia physical status; data are shown as median (Q_1_ = first quartile; Q_3_ = third quartile); * *p* < 0.05.

**Table 2 ijms-21-00084-t002:** Results of laboratory tests of preoperative evaluation.

Parameter	GA Group	SA Group	*p-*Value
MAP [mm Hg]	130 (120; 143)	132 (120; 140)	0.47
Gly [mmol/l]	5.55 (5.33; 5.78)	5.7 (5.18; 6.08)	0.29
Na [mmol/l]	139 (137.8; 140.5)	140 (139; 142)	0.53
K [mmol/l]	4.2 (4; 4.5)	4.6 (4.58; 4.8)	0.54
Cl [mmol/l]	103 (100; 105)	104 (102; 107)	0.27
Anest.	180 (120; 273)	107.5 (91.3; 118.8)	0.001 *
Duration [min]			

MAP—mean arterial pressure; Gly—glycemia; Na—natremia; K—kalemia; Cl—chloremia; Anest. duration—duration of anesthesia.; data are shown as median (Q_1_ = first quartile; Q_3_ = third quartile); * *p* < 0.05.

**Table 3 ijms-21-00084-t003:** Results of DNA damage in general and spinal anesthesia groups.

	GA		SA	
	Before	After	Before	After
SSB	7.49	10.05 **	4.00	4.18
	(5.09; 9.66)	(6.97; 11.63)	(1.71; 8.80)	(1.91; 7.39)
ENDO III	8.65	11.85 **	5.83	6.60
	(6.12; 10.20)	(8.27; 13.47)	(3.02; 12.78)	(3.42; 12.73)
FPG	7.97	11.75 **	6.72	8.61
	(5.72; 12.04)	(8.38; 15.32)	(3.34; 15.77)	(4.76; 15.60)

Values are expressed as median (Q_1_ = first quartile; Q_3_ = third quartile); ** *p* < 0.0001.

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
