# Peer review of "Does Neuraxial Anesthesia as General Anesthesia Damage DNA? A Pilot Study in Patients Undergoing Orthopedic Traumatological Surgery"

_ijms, 2019, doi:10.3390/ijms21010084_

Round 1

Reviewer 1 Report

In this manuscript by Kucharova et al, a pilot study on levels of DNA damage due to exposure to general (GA) and neuraxial spinal anesthesia (SA) is presented. The study is based on two groups, each consisting of 20 patients, undergoing orthopedic traumatological surgery. The percentage of cells displaying unrepaired DNA damage, of different types, was evaluated by the comet assay method, looking at individual cells. The results point to a potential difference in degree of DNA damage induced by the two different anesthesia protocols.

The aim of this study is indeed valuable. Being able to use a method for sufficient pain control during surgery, without the risk of causing irreversible DNA damage in the patient, thereby potentially causing development of malignancies, is naturally of significant importance.

The manuscript is in general well written and the introduction gives a comprehensive and adequate summary of the background for the study.

However, although this study presents a statistically significant difference between levels of DNA damage after exposure to GA and SA there are concerns to be raised.

It is clear from the abstract and the discussion that this is a pilot study, potentially performed to motivate a bigger study. I think that this should also be clear from the title, for readers to perceive the content and the message correctly.

I think that it is far too strong to use the word “proved” in the following statement by the authors; It was proved that general anesthesia causes statistically significant damage to the DNA of patients, whereas neuraxial anesthesia had no negative influence.

This is based on that the number of patients are very small and also the fact that there is a significant difference in age, and in time for surgery procedure. This makes it hard to judge whether the difference in DNA damage frequency is indeed different based on the method for anesthesia. Also, and potentially more important, the level of DNA damage pre-surgery is higher in the group receiving GA, compared with the group receiving SA. Is this difference significant? It seems to be at least as big as the difference before and after surgery in the group receiving GA. This needs to be analyzed and at least discussed. What is the pre-surgery history of the two groups? at least include in the discussion the fact that they have not been treated with chemo or radiation therapy during the last twelve months. If anything is known about the actual differences between the groups that would also be informative.

In the introduction it is indicated that the primary goal of the study is to evaluate the comet assay – how was this done? Was it done? What was done to evaluate the correctness of the measurement of the different types of damages? An evaluation of damages induced in vitro of known types, including positive and negative controls, should be done to be able to say that an evaluation has been performed. At least if any evaluation was done this should be described and discussed.

For Figure 1 and 2:

It is not very clear how the DNA damage before and after exposure to GA, can be significantly different statistically, considering the big variation within the two sets of samples and the overlapping error bars between the before and after samples.

I think that Box-and-whisker plots, with the individual values for each patient, would show more clearly display the spreading within the groups, if one or two individuals are outliers or whether there is an even distribution within the group.

Discuss or describe the statistics used in more detail. Why is the Wilcoxon method motivated and best for this set of data?

Discuss how a larger study should be best performed to secure that not only possible statistical differences but also biological differences can be detected. Ideally a positive and a negative control should be included. As a negative control one group of patients/individuals could be donating blood with the same time interval only without being exposed to the treatment. I assume that the GA group is a type of positive control, since DNA damage has been shown to induced before.

Author Response

Dear Reviewer,

thank you for your comments that help to improve the quality of this manuscript. It was very useful for us. We have carefully processed every note. We resubmitted repaired and completed manuscript.

Reviewer 2 Report

COMMENTS TO THE AUTHOR(S)

The authors designed and executed an excellent study aimed to evaluate effects of anesthesia on DNA damage, which was verified by the comet assay. The results indicated that DNA damage rate in lymphocyte cells isolated from patients underwent general anesthesia was higher than that from patients underwent neuraxial anesthesia. The topic is innovative and the methodology is scientifically sound, although there are many things to be elucidated.

The authors need to better explain how lymphocyte cells uptake anesthesia and what is the target molecules for induction of DNA damage in lymphocyte cells in the introduction and/or discussion sections. Finally, it is unclear how the findings from this work would be transferrable to other cases or inspire/guide other scientists. I think this article could be published in Int J Mol Sci, however, I suggest minor revision before the editor(s) decide the final decision.

Some minor points:

Line 45: Abbreviations GABA and NMDA require formal names.

Line 100: “that prove that” can be changed to “proved that”

Figures: The authors should align decimal places. (ex. 15% vs 15.00%)

Line 159: The authors should show the results of dependence between duration and SSB in the SA group in addition to those in the GA group.

Author Response

Dear reviewer,

thank you for your comments that help to improve the quality of this manuscript. It was very useful for us. We have carefully processed every note. We corrected minor points and tried to answer to interesting question. We resubmitted repaired and completed manuscript.